# Group-Based Trajectory Modeling of N-Terminal Pro-Brain Natriuretic Peptide Levels in Pulmonary Artery Hypertension Associated with Connective Tissue Disease

**DOI:** 10.3390/healthcare12161633

**Published:** 2024-08-16

**Authors:** Heng Tang, Fengyun Lu, Yingheng Huang, Qiang Wang, Xiaoxuan Sun, Miaojia Zhang, Lei Zhou

**Affiliations:** 1Department of Cardiology, First Affiliated Hospital of Nanjing Medical University, Nanjing 210029, China; tangheng49@163.com; 2Department of Rheumatology, First Affiliated Hospital of Nanjing Medical University, Nanjing 210029, China; lufengyun@jsph.org.cn (F.L.); huangyingheng123@163.com (Y.H.); wangqiang@jsph.org.cn (Q.W.); sunxiaoxuan@jsph.org.cn (X.S.)

**Keywords:** connective tissue disease, pulmonary artery hypertension, N-terminal pro-brain natriuretic peptide, group-based trajectory modeling

## Abstract

Group-based trajectory modeling (GBTM) allows the trajectory analyses of repeated N-terminal pro-brain natriuretic peptide (NT-proBNP) measurements during follow-up visits of pulmonary artery hypertension associated with connective tissue disease (CTD-PAH) patients. This study aimed to (1) identify trajectories of NT-proBNP changing over time, (2) explore the association between NT-proBNP trajectories and prognosis, and (3) explore the effects of baseline clinical characteristics on NT-proBNP trajectories. A retrospective, single-centred, observational study was performed on 52 CTD-PAH patients who had undergone at least three follow-up visits within 1 year from baseline. Four NT-proBNP trajectories were identified using GBTM: low stability (*n* = 15, 28.85%), early remission (remission within 3 months) (*n* = 20, 38.46%), delayed remission (remission after 6 or 9 months) (*n* = 11, 21.15%), and high stability (*n* = 6, 11.54%). The low-stability and early-remission trajectories were related to a similar positive prognosis, while the delayed-remission and high-stability trajectories were associated with a gradually worsening prognosis (*p* = 0.000). Intensive CTD immunotherapy (corticosteroids plus immunosuppressants) was the only factor that remained significant after least absolute shrinkage and selection operator regression and multivariate logistic regression, and was independently associated with a lower risk NT-proBNP trajectory (*p* = 0.048, odds ratio = 0.027, 95% confidence interval: 0.001–0.963), which preliminarily indicated a benefit of CTD-PAH patients undergoing intensive CTD immunotherapy.

## 1. Introduction

Pulmonary hypertension (PH) is a fatal pathophysiological disorder defined by a mean pulmonary arterial pressure (mPAP) > 20 mmHg at rest measured by right heart catheterization (RHC), and can be caused by a variety of clinical conditions [1]. Pulmonary artery hypertension (PAH) is hemodynamically characterized by pre-capillary PH in the absence of other causes of pre-capillary PH such as PH associated with lung diseases and PH associated with pulmonary artery obstructions, which meets the following criteria: mPAP > 20 mmHg, pulmonary arterial wedge pressure (PAWP) ≤ 15 mmHg, and pulmonary vascular resistance (PVR) > 2 Wood units (WU) [2]. Among all of the clinical subtypes of PAH, PAH associated with connective tissue disease (CTD-PAH) is the second most prevalent subtype after idiopathic PAH (IPAH), accounting for 15–30% of PAH cases [3,4].

Although considerable effort has been made to identify novel biomarkers of PAH diagnosis, classification, treatment response, and prognosis prediction, brain natriuretic peptide (BNP) and N-terminal pro-BNP (NT-proBNP), which reflect the volume load of the ventricles, remain the only biomarkers routinely used in clinical practice, and are included in almost all of the existing risk assessment scales of PAH, such as the 2018 World Symposium on Pulmonary Hypertension (WSPH) simplified risk assessment scale [5], the 2022 European Society of Cardiology (ESC)/European Respiratory Society (ERS) guidelines updated risk stratification tool [2], the Registry to Evaluate Early and Long-Term PAH Disease Management (REVEAL) 2.0 score [6], and the REVEAL Lite 2.0 score [7]. These risk assessment scales are based on a combination of single measurements of parameters at a certain time point; however, trajectories of parameters changing over time obtained from repeated measurements can provide more information for the response of PAH patients to treatment and their long-term prognosis. Group-based trajectory modeling (GBTM) is a statistical application of finite mixture modeling on longitudinal observational data that assign patients who follow similar trajectories to distinct subgroups, and has been widely used in clinical data analyses [8,9].

Therefore, to identify trajectories of NT-proBNP changing over time, we applied GBTM to repeated measurements of NT-proBNP levels in peripheral blood within 1 year from baseline in a CTD-PAH cohort, and we subsequently evaluated the association between NT-proBNP trajectories and prognosis. To find out the potential protective factors or risk factors of distinct NT-proBNP trajectories, we further explored the effects of baseline clinical characteristics on NT-proBNP trajectories (Figure 1).

## 2. Materials and Methods

### 2.1. Study Design and Population

This was a retrospective, single-centred, observational study of 52 CTD-PAH patients who had undergone at least three follow-up visits within 1 year from baseline at First Affiliated Hospital of Nanjing Medical University, Nanjing, China, from 2017 to 2023. Patients who fulfilled all of the inclusion criteria were recruited: (1) patients with a diagnosis of PAH, (2) patients with a primary connective tissue disease (CTD), and (3) patients who had undergone at least three follow-up visits within 1 year from baseline. A positive diagnosis of PAH was confirmed by RHC which met the following criteria: mPAP > 20 mmHg, PAWP ≤ 15 mmHg, and PVR > 2 WU. Positive diagnoses of CTDs were based on authoritative criteria: the 2019 America College of Rheumatology (ACR)/European League Against Rheumatism (EULAR) classification criteria for systemic lupus erythematosus (SLE) [10], the 2002 international classification criteria for primary Sjogren’s syndrome (pSS) [11], the 2013 ACR/EULAR classification criteria for systemic sclerosis (SSc) [12], the 2010 ACR/EULAR classification criteria for rheumatoid arthritis (RA) [13], and the 1972 Sharp criteria for mixed connective tissue disease (MCTD) [14]. Patients presenting with clinical and serological features suggestive of systemic autoimmune diseases but not fulfilling criteria for any defined CTD were defined as having undifferentiated connective tissue disease (UCTD). Exclusion criteria were as follows: (1) patients with group 2 PH associated with left heart disease, (2) patients with group 3 PH associated with lung diseases and/or hypoxia, (3) patients with group 4 PH associated with pulmonary artery obstructions, (4) patients with group 5 PH with unclear and/or multifactorial mechanisms, (5) patients with group 1 PAH except for CTD-PAH [2], (5) patients aged over 75 years or below 18 years, (6) patients with an estimated glomerular filtration rate (eGFR) < 60 mL/min, (7) patients who had not undergone at least three follow-up visits within 1 year from baseline, (8) patients missing NT-proBNP measurements at baseline or in follow-up visits, and (9) patients missing RHC or echocardiography data at baseline. The primary end point was defined as the following: (1) all-cause death, or (2) re-hospitalization for worsening PAH (any re-hospitalization for worsening PAH, lung transplantation or heart-lung transplantation, atrial septostomy, or initiation of parenteral prostanoid therapy). Our study was approved by the Medical Ethics Committee of First Affiliated Hospital of Nanjing Medical University. The Clinical Trial ID for our study is NCT05980728.

### 2.2. Data Collection

The baseline of CTD-PAH in this study was defined as the date of the first RHC when PAH was diagnosed. Baseline data of demographics, detailed case histories, 6-minute walk distance (6MWD), laboratory examinations, echocardiography, and RHC were obtained from electronic records systems and specialist services databases by clinicians blinded to the patient outcomes. Activities of primary CTDs were evaluated by experienced rheumatologists blinded to the patient outcomes with the following assessment scores: the SLE activity index (SLEDAI) for SLE [15], the EULAR pSS disease activity index (ESSDAI) for pSS [16], the modified Rodnan skin score (mRSS) for SSc [17], and the disease activity score with 28 joint counts (DAS28) for RA [18]. The 2018 WSPH simplified risk assessment scale risk groups were classified by experienced rheumatologists blinded to the patient outcomes according to the data collected. Patients were followed up every 3 months in the outpatient clinic for re-evaluation including measurements of NT-proBNP levels in peripheral blood.

### 2.3. Latent Trajectory Modeling

We applied GBTM to the longitudinal observational data of NT-proBNP to identify distinct NT-proBNP trajectories. GBTM was conducted using SAS 9.4 (traj procedure). We generated models in which the number of trajectories ranged from three to four, and the shape of the trajectories was represented by polynomial functions ranging from zero to third order. The appropriate model was determined using the following criteria: (1) the fitting of each trajectory was statistically significant, (2) the average posterior probability (APP) of each trajectory was > 0.7, (3) the odds of correct classification (OCC) of each trajectory was > 5, (4) the relative entropy (Ej) was > 0.5, and (5) the number of individuals assigned to each trajectory exceeded 3% of the total population. Of the models that met all of the criteria (Appendix A), the final model was chosen based on its relatively small Bayesian information criterion (BIC) and clinical interpretability [8,9]. GBTM was conducted blinded to the baseline and prognostic data.

### 2.4. Statistics

Non-normally distributed variables are described by the median and interquartile range, normally distributed variables are described by the mean and standard deviation, and categorical variables are described as number (%). Comparisons of two groups of normally distributed variables were performed using independent sample *t* tests. Comparisons of two groups of non-normally distributed variables were performed using Mann–Whitney U tests. Comparisons of proportions were performed using chi-square or Fisher’s exact tests.

The association between NT-proBNP trajectories and prognosis was evaluated using Kaplan–Meier curves with log-rank tests. To explore the effects of baseline clinical characteristics on NT-proBNP trajectories, least absolute shrinkage and selection operator (LASSO) regression and logistic regression were used. The extraction of variables was performed based on LASSO regression (glmnet package). The optimal variable combination was the variable corresponding to the highest ln(lambda) value that resulted in a misclassification error within one standard error of the minimum cross-validated error [19]. These selected variables were subsequently applied to multivariate logistic regression for further validation. 

All of the statistics except for GBTM were completed using R 4.3.2 and IBM SPSS Statistics version 25. For all of the tests, a *p*-value < 0.05 was considered to be statistically significant.

## 3. Results

### 3.1. Patient Characteristics

According to the inclusion and exclusion criteria, 52 CTD-PAH patients were eligible and recruited. Clinical details of the 52 CTD-PAH patients are summarized in Appendix A, including 22 (42.30%) SLE, 13 (25.00%) pSS, six (11.54%) SSc, two (3.85%) RA, three (5.77%) MCTD, and six (11.54%) UCTD patients. All of the participants were female (100.00%), with a median age of 41.00 (IQR 34.00, 55.25) years. The median CTD duration was 0.00 (IQR 0.00, 6.50) years, and 44.23% patients were suffering with active CTDs at baseline. More than half of the patients (75.00%) received intensive CTD immunotherapy (glucocorticoids plus immunosuppressants) for primary CTDs. The median PAH duration was 1.00 (IQR 0.00, 2.00) years. According to the 2018 WSPH simplified risk assessment scale, 18 (34.62%) patients were at a low risk, 24 (46.15%) patients were at an intermediate risk, and 10 (19.23%) patients were at a high risk. All of the CTD-PAH patients were under standard PAH-targeted drug therapy and were treated with at least one targeted drug. Six (11.54%) patients received monotherapy, half of them were treated with endothelin receptor antagonists (ERA) (ambrisentan, macitentan, or bosentan), and the rest were treated with phosphodiesterase 5 inhibitors (PDE5i) (Sildenafil or Tadalafil). Thirty-two (61.54%) patients received dual therapy, among which, 30 patients were treated with ERA plus PDE5i, one patient was treated with ERA plus prostacyclin receptor agonist (PRA) (Selexipag), and one patient was treated with PDE5i plus PRA. Fourteen (26.92%) patients received triple therapy, which was a combination of ERA, PDE5i, and PRA. 

### 3.2. NT-proBNP Trajectories

GBTM was used to identify trajectories of NT-proBNP changing over time. For the reason that absolute values of NT-proBNP could reach extremely high levels, and thus slight changes would be ignored, we converted absolute values of NT-proBNP into ranked data that indicated low risk, intermediate risk, and high risk according to thresholds in the 2022 ESC/ERS guidelines’ updated risk stratification tool: <300 pg/mL, 300–1100 pg/mL, and >1100 pg/mL [2]. A four-grouped cubic model was selected and the orders of polynomials were eventually reduced to 0320, which met all of the criteria with a relatively small BIC and clinical interpretability (Appendix A, Figure 2g). The four NT-proBNP trajectories were as follows: low stability (*n* = 15, 28.85%), with NT-proBNP persistently maintained at a low risk; early remission (*n* = 20, 38.46%), with an early remission of NT-proBNP to a low risk within 3 months from baseline; delayed remission (*n* = 11, 21.15%), with a delayed remission of NT-proBNP to a low risk after 6 or 9 months from baseline; and high stability (*n* = 6, 11.54%), with NT-proBNP persistently maintained at a high risk (Figure 2a–f).

### 3.3. Association between NT-proBNP Trajectories and Prognosis

During a median of 21 months of follow-up assessments, six (11.54%) patients experienced all-cause death or re-hospitalization for worsening PAH. Kaplan–Meier curves of the four NT-proBNP trajectories are shown in Figure 3, which illustrate significant differences in the prognosis (*p* = 0.000). The low-stability trajectory and the early-remission trajectory were related to a similar positive prognosis (no all-cause death or re-hospitalization for worsening PAH occurred), while the occurrence of all-cause death or re-hospitalization for worsening PAH increased sharply in patients with a delayed-remission trajectory, indicating that an early remission of NT-proBNP within 3 months from baseline, but not a delayed remission of NT-proBNP after 6 or 9 months from baseline, could improve the prognosis. The high-stability trajectory was associated with the worst prognosis among the four trajectories, which was not surprising according to clinical experience.

### 3.4. Effects of Baseline Clinical Characteristics on NT-proBNP Trajectories

The baseline clinical characteristics of patients with each NT-proBNP trajectory are described in Appendix A. The early-remission, delayed-remission, and high-stability trajectories all began with high NT-proBNP levels; however, the early-remission trajectory was related to a positive prognosis, while the delayed remission and high-stability trajectories were associated with a poor prognosis. Therefore, comparisons and regressions were conducted between the early remission trajectory and a combination of the delayed-remission and high-stability trajectories.

Comparisons of baseline clinical characteristics revealed that age (*p* = 0.006), CTD duration (*p* = 0.024), intensive CTD immunotherapy (*p* = 0.005), 6MWD (*p* = 0.004), World Health Organization functional class (WHO-FC) (*p* = 0.023), and echocardiography parameters, including right atrial diameter (RAD) (*p* = 0.010), right ventricular end diastolic diameter (RVDd) (*p* = 0.012), and pericardial effusion (*p* = 0.012) significantly differed (Table 1). Distribution of baseline WSPH groups across NT-proBNP trajectories is shown in Appendix A (*p* = 0.000). The low-stability trajectory mostly consisted of the WSPH ‘low risk’ group, while the other trajectories mostly consisted of WSPH ‘intermediate risk’ and ‘high risk’ groups. The proportion of WSPH ‘high risk’ group in the delayed remission and high-stability trajectories was dramatically higher than in the early-remission trajectory.

After extracting variables based on LASSO regression, age, intensive CTD immunotherapy, WHO-FC, RVDd, and pericardial effusion were retained and applied to multivariate logistic regression (Table 2). Intensive CTD immunotherapy was the only factor that remained significant after multivariate logistic regression, and thus was confirmed to be independently associated with a lower risk NT-proBNP trajectory (*p* = 0.048, odds ratio (OR) = 0.027, 95% confidence interval (CI): 0.001–0.963) (Table 2), which indicated a positive response of CTD-PAH patients to intensive CTD immunotherapy partly shown by an earlier NT-proBNP remission.

## 4. Discussion

In this study, we applied GBTM-based trajectory analyses to repeated measurements of NT-proBNP within 1 year from baseline in a clinically, experimentally, and hemodynamically-confirmed CTD-PAH cohort, and we subsequently evaluated the association between NT-proBNP trajectories and prognosis. Four NT-proBNP trajectories were identified. The low-stability trajectory was characterized by a maintenance of NT-proBNP at a low risk during the whole follow-up assessment. The early-remission, delayed-remission, and high-stability trajectories all began with high NT-proBNP levels. Patients with an early-remission trajectory reached an NT-proBNP remission within 3 months from baseline. Patients with a delayed-remission trajectory also reached an NT-proBNP remission eventually, but the remission took 6 to 9 months. Patients with a high-stability trajectory failed to reach an NT-proBNP remission until the end of the follow-up assessments. The early-remission, delayed-remission, and high-stability trajectories described a deteriorating remission of NT-proBNP, and were associated with an increased occurrence of all-cause death or re-hospitalization for worsening PAH. A register-based observational study of 383 PAH patients by Klyhammar et al. revealed that the survival was similar for patients who remained at or improved to a low risk (evaluated by the 2015 ESC/ERS guidelines risk stratification tool) after a median of 4 months of follow-up assessments, and the results were unchanged in the subgroups of patients with IPAH or CTD-PAH [20]. These results were in accordance with the results in our cohort that the prognosis was similar for CTD-PAH patients with the low-stability and early-remission NT-proBNP trajectories, of which the NT-proBNP levels remained at or improved to a low risk after 3 months of follow-up assessments. However, the effects of the time to achieve a low risk on the prognosis of PAH patients were not explored in their study. In our cohort, we found that an early remission of NT-proBNP within 3 months from baseline could improve the prognosis, while a delayed remission of NT-proBNP after 6 or 9 months from baseline could not, which demonstrated the key role of the time to achieve NT-proBNP remission in the CTD-PAH prognosis prediction. These findings emphasized the necessity of repeated measurements and trajectory analyses of NT-proBNP in the management of CTD-PAH patients, especially a re-measurement at 3 months from baseline for patients with high-baseline NT-proBNP levels.

We further explored the effects of baseline clinical characteristics on NT-proBNP trajectories. Potential predictors include age (gender was not included for the reason that all of the participants were female), CTD type, CTD duration, CTD activity, CTD immunotherapy, PAH duration, PAH-targeted drug therapy, and PAH severity, which are reflected by 6MWD, WHO-FC, echocardiography parameters, and RHC parameters. Comparisons between differential NT-proBNP remission trajectories revealed that age, CTD duration, CTD immunotherapy, and PAH severity (6MWD, WHO-FC, RAD, RVDd, and pericardial effusion) significantly differed. The relationship between baseline PAH severity and NT-proBNP trajectories was further described by the distribution of baseline WSPH groups across the trajectories. Patients classified into higher-risk WSPH groups at baseline were more likely to develop a higher-risk NT-proBNP trajectory. These 21 potential predictors were applied to LASSO regression for variables extraction, and five of them, including age, intensive CTD immunotherapy, WHO-FC, RVDd, and pericardial effusion, were retained for multivariate logistic regression. Intensive CTD immunotherapy (corticosteroids plus immunosuppressants) was the only factor that remained significant after multivariate logistic regression, and thus was confirmed to be independently associated with an earlier remission of NT-proBNP. The other four variables including age, WHO-FC, RVDd, and pericardial effusion failed to remain significant after being adjusted with all of the covariates. Although there were associations between baseline PAH severity and NT-proBNP trajectories, WHO-FC, RVDd, and pericardial effusion were not significant enough in the multivariate logistic regression for the possible reason that they only reflected partial dimensions of baseline PAH severity. Notably, PAH target drug therapy was not included in the multivariate logistic regression for the reason that it was eliminated by LASSO regression; it should be mentioned that all of the CTD-PAH patients in our cohort were undergoing standard PAH target drug therapy, indicating little interference from inappropriate PAH target drug therapy to the results. Although strategies for using PAH-targeted drugs have been well described in authoritative guidelines, strategies for using CTD immunotherapy in CTD-PAH have been rarely mentioned [2]. Only a few studies have investigated the effectiveness of the single or combined use of glucocorticoids and immunosuppressants in CTD-PAH. To date, there was only one large cohort study of more than 100 CTD-PAH patients focused on this. In the Chinese SLE treatment and research group (CSTAR)-PAH study, which was performed in a large multi-centred cohort of 310 SLE-associated PAH patients, it was reported that patients under intensive immunotherapy were more likely to reach treatment goals in a subgroup of patients with serositis at baseline, but not in the overall cohort [21]. A recent systematic review summarized the results of studies on the effectiveness of CTD immunotherapy in CTD-PAH, including seven cohorts, one trial, and one case series for a total of 439 patients, which supported using intensive CTD immunotherapy in CTD-PAH, particularly in SLE-associated PAH [22], which was not surprising for the reason that the CSTAR-PAH study was the major study in this systematic review. Although the end point and the patient population were different, effects of intensive CTD immunotherapy were both positive in their studies and ours, which preliminarily indicated a benefit of CTD-PAH patients undergoing intensive CTD immunotherapy.

The main limitation of the study was potential selection bias in a retrospective, single-centred study with a relatively small sample size partly due to the low prevalence of PAH. It should be noted that all of the well-diagnosed CTD-PAH patients who had undergone at least three follow-up visits within 1 year from baseline in our department from 2017 to 2023 were recruited in the study, indicating that our cohort is a ‘real-world’ cohort in clinical practice. The second major limitation was that the number of patients with SSc, RA, MCTD, and UCTD included in the CTD-PAH cohort was too small for conducting subgroup analyses, which would be goals of our future studies. Intensive CTD immunotherapy was confirmed to result in an earlier NT-proBNP remission in our cohort, which provided evidence for establishing strategies for using CTD immunotherapy in CTD-PAH. It is essential to determine if our findings are generalizable to other cohorts of CTD-PAH patients; therefore, well-designed multi-centred randomized controlled trials with large sample sizes are needed.

## 5. Conclusions

Four GBTM-based NT-proBNP trajectories were identified that described a deteriorating remission of NT-proBNP, and were associated with an increased occurrence of all-cause death or re-hospitalization for worsening PAH. We found that an early remission of NT-proBNP within 3 months from baseline could improve the prognosis of CTD-PAH patients, while a delayed remission of NT-proBNP after 6 or 9 months from baseline could not, which demonstrated the key role of the time to achieve NT-proBNP remission in CTD-PAH prognosis prediction and the necessity of repeated measurements and trajectory analyses of NT-proBNP in the management of CTD-PAH patients. We found that intensive CTD immunotherapy was a protective factor related to an earlier remission of NT-proBNP, which preliminarily indicated a benefit of CTD-PAH patients undergoing intensive CTD immunotherapy.

## Figures and Tables

**Figure 1 healthcare-12-01633-f001:**
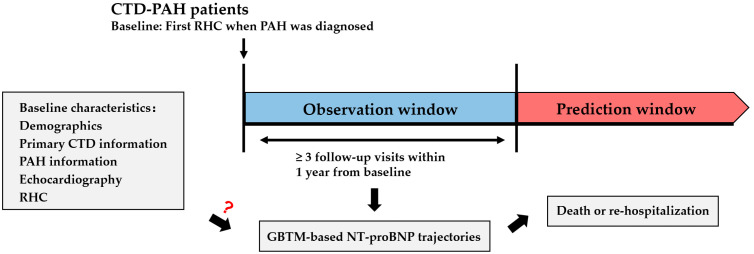
Study design. Abbreviations: CTD, connective tissue disease; PAH, pulmonary artery hypertension; RHC, right heart catheterization; GBTM, group-based trajectory modeling; NT-proBNP, N-terminal pro-brain natriuretic peptide.

**Figure 2 healthcare-12-01633-f002:**
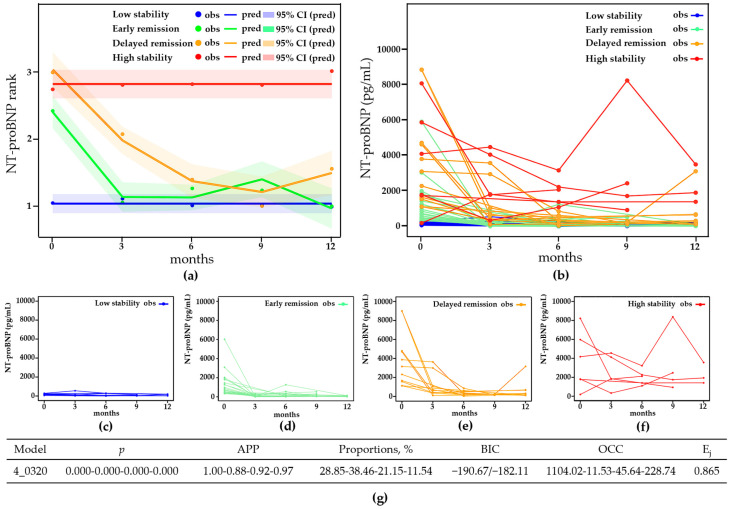
GBTM-based NT-proBNP trajectories and model evaluation. (**a**) Trajectories of NT-proBNP ranks (1: <300 pg/mL, 2: 300–1100 pg/mL, 3: >1100 pg/mL). (**b**) Absolute values of NT-proBNP in the four trajectories: (**c**) the low-stability trajectory, (**d**) the early-remission trajectory, (**e**) the delayed-remission trajectory, and (**f**) the high-stability trajectory. (**g**) Parameters of the model. Abbreviations: NT-proBNP, N-terminal pro-brain natriuretic peptide; CI, confidence interval; APP, average posterior probability; BIC, Bayesian information criterion; OCC, odds of correct classification; Ej, relative entropy; obs, observed values; pred, predictive values.

**Figure 3 healthcare-12-01633-f003:**
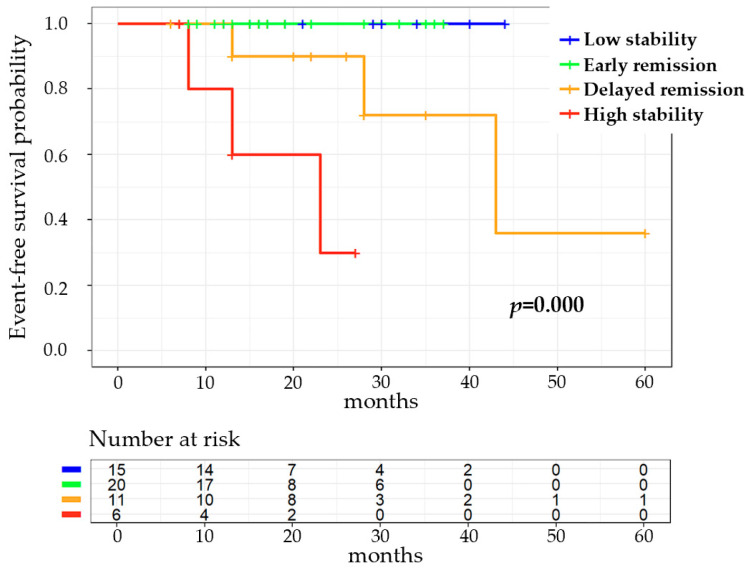
Association between NT-proBNP trajectories and prognosis.

**Table 1 healthcare-12-01633-t001:** Baseline clinical characteristics of the early-remission trajectory and a combination of the delayed-remission and high-stability trajectories.

Baseline Clinical Characteristics	NT-proBNP Trajectories		*p*
	Early Remission (*n* = 20)	Delayed Remission and High Stability (*n* = 17)	
Age, years	34.00 (25.00, 45.00)	48.00 (36.50, 57.50)	0.006
Primary CTD			0.120
SLE, *n* (%)	8 (40.00)	6 (35.30)	
pSS, *n* (%)	5 (25.00)	4 (23.53)	
SSc, *n* (%)	0 (0.00)	5 (29.41)	
RA, *n* (%)	1 (5.00)	0 (0.00)	
MCTD, *n* (%)	2 (10.00)	1 (5.88)	
UCTD, *n* (%)	4 (20.00)	1 (5.88)	
CTD duration, years	0.00 (0.00, 0.00)	2.00 (0.00, 7.50)	0.024
Active CTD, *n* (%)	11 (55.00)	6 (35.29)	0.231
Intensive CTD immunotherapy, *n* (%)	19 (95.00)	9 (52.94)	0.005
PAH duration, years	0.00 (0.00, 1.00)	1.00 (0.00, 2.00)	0.242
WSPH groups			0.033
Low risk, *n* (%)	5 (25.00)	1 (5.88)	
Intermediate risk, *n* (%)	13 (65.00)	8 (47.06)	
High risk, *n* (%)	2 (10.00)	8 (47.06)	
6MWD, m	458.50 (392.75, 538.75)	368.00 (227.00, 443.50)	0.004
WHO-FC			0.023
I, *n* (%)	1 (5.00)	0 (0.00)	
II, *n* (%)	10 (50.00)	3 (17.65)	
III, *n* (%)	9 (45.00)	11 (64.71)	
IV, *n* (%)	0 (0.00)	3 (17.65)	
RAD, mm	40.00 (38.25, 44.75)	46.00 (41.50, 51.00)	0.010
RVDd, mm	41.80 ± 4.07	46.12 ± 5.78	0.012
TRV, cm/s	401.00 ± 72.25	424.29 ± 62.73	0.307
TAPSE/PASP ratio	0.27 ± 0.12	0.20 ± 0.09	0.061
Pericardial effusion, *n* (%)	7 (35.00)	13 (76.47)	0.012
mPAP, mmHg	41.00 (34.25, 57.75)	49.00 (39.50, 62.00)	0.131
SVO_2_, %	63.50 (59.25, 67.75)	57.00 (46.00, 70.50)	0.293
mRAP, mmHg	5.50 (3.00, 9.00)	7.00 (4.50, 9.00)	0.340
PAWP, mmHg	8.20 ± 3.02	9.12 ± 3.08	0.368
mRVP, mmHg	22.50 (19.50, 34.75)	28.00 (24.50, 37.00)	0.184
PVR, Wood	8.20 (5.55, 13.13)	10.63 (6.63, 21.88)	0.170
Cardiac index, L/min/m^2^	2.72 ± 0.75	2.51 ± 0.89	0.424
PAH-targeted drug therapy			0.439
ERA, *n* (%)	0 (0.00)	0 (0.00)	
PDE5i, *n* (%)	0 (0.00)	1 (5.88)	
ERA + PDE5i, *n* (%)	12 (60.00)	10 (58.82)	
ERA + PRA, *n* (%)	0 (0.00)	1 (5.88)	
PDE5i + PRA, *n* (%)	0 (0.00)	0 (0.00)	
ERA + PDE5i + PRA, *n* (%)	8 (40.00)	5 (29.41)	

Abbreviations: SLE, systemic lupus erythematosus; pSS, primary Sjogren’s syndrome; SSc, systemic sclerosis; RA, rheumatoid arthritis; MCTD, mixed connective tissue disease; UCTD, undifferentiated connective tissue disease; CTD, connective tissue disease; PAH, pulmonary artery hypertension; WSPH, World Symposium on Pulmonary Hypertension; 6MWD, 6-minute walk distance; WHO-FC, World Health Organization functional class; RAD, right atrial diameter; RVDd, right ventricular end diastolic diameter; TRV, tricuspid regurgitation velocity; TAPSE, tricuspid annular plane systolic excursion; PASP, pulmonary artery systolic pressure; mPAP, mean pulmonary artery pressure; SVO_2_, oxygen saturation of mixed venous blood; mRAP, mean right atrial pressure; PAWP, pulmonary artery wedge pressure; mRVP, mean right ventricular pressure; PVR, pulmonary vascular resistance; ERA, endothelin receptor antagonists; PDE5i, phosphodiesterase 5 inhibitors; PRA, prostacyclin receptor agonist.

**Table 2 healthcare-12-01633-t002:** Effects of baseline clinical characteristics on NT-proBNP trajectories.

Baseline Clinical Characteristics	LASSO Regression Coefficient	Univariate Logistic Regression		Multivariate Logistic Regression	
		*p*	OR (95% CI)	*p*	OR (95% CI)
Age	0.014	0.015	1.080 (1.015–1.149)	0.061	1.106 (0.995–1.229)
Primary CTD		>0.05			
CTD duration		0.259	1.062 (0.957–1.179)		
Active CTD		0.234	0.446 (0.118–1.685)		
Intensive CTD immunotherapy	−0.718	0.013	0.059 (0.006–0.548)	0.048	0.027 (0.001–0.963)
PAH duration		0.893	0.985 (0.787–1.232)		
6MWD rank		0.024	2.679 (1.136–6.319)		
WHO-FC	0.307	0.013	5.689 (1.448–22.348)	0.077	7.710 (0.799–74.377)
RAD		0.023	1.176 (1.022–1.353)		
RVDd	0.029	0.020	1.203 (1.029–1.406)	0.205	1.171 (0.918–1.494)
TRV		0.299	1.005 (0.995–1.015)		
TAPSE/PASP ratio rank		0.056	2.402 (0.976–5.908)		
Pericardial effusion	0.319	0.015	6.036 (1.417–25.710)	0.054	15.887 (0.955–264.229)
mPAP		0.139	1.035 (0.989–1.084)		
SVO_2_		0.110	0.946 (0.885–1.013)		
mRAP		0.489	1.069 (0.884–1.294)		
PAWP		0.358	1.109 (0.889–1.383)		
mRVP		0.573	1.019 (0.954–1.088)		
PVR		0.119	1.074 (0.982–1.176)		
Cardiac index		0.413	0.709 (0.311–1.616)		
PAH-targeted drug therapy		>0.05			

Abbreviations: CTD, connective tissue disease; PAH, pulmonary artery hypertension; 6MWD, 6-minute walk distance; WHO-FC, World Health Organization functional class; RAD, right atrial diameter; RVDd, right ventricular end diastolic diameter; TRV, tricuspid regurgitation velocity; TAPSE, tricuspid annular plane systolic excursion; PASP, pulmonary artery systolic pressure; mPAP, mean pulmonary artery pressure; SVO_2_, oxygen saturation of mixed venous blood; mRAP, mean right atrial pressure; PAWP, pulmonary artery wedge pressure; mRVP, mean right ventricular pressure; PVR, pulmonary vascular resistance.

## Data Availability

The data are not publicly available due to ethical restrictions.

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
