# Peer review of "Group-Based Trajectory Modeling of N-Terminal Pro-Brain Natriuretic Peptide Levels in Pulmonary Artery Hypertension Associated with Connective Tissue Disease"

_healthcare, 2024, doi:10.3390/healthcare12161633_

Round 1
Reviewer 1 Report
Comments and Suggestions for Authors
Very well written. Interesting topic for presentation and discussion. They have recognized one of the drawbacks of the study as low patient count and only females were part of the study. Further research and retrospective trials are needed to confirm the data presented in the study.
It looks like all the patients received PAH treatment. In the supplement they have mentioned monotherapy, dual therapy and triple therapy for PAH. They did not mention what kind of medications they received. In the world of PAH with limited treatment options, it will be interesting to know what they have received and if that played in any part in the 4 trajectories.
In the supplement table, its written that Intensive CTD immunotherapy was given in 39 out of the 52 patients. If you add up all patients in 4 groups its coming to 46 (11+19+7+9 = 46).
Author Response
Thank you very much for taking the time to review this manuscript. Please find the detailed responses below and the corresponding revisions highlighted in the re-submitted files.
Comments 1: They have recognized one of the drawbacks of the study as low patient count and only females were part of the study. Further research and retrospective trials are needed to confirm the data presented in the study.
Response 1: Thank you very much for your understanding. CTD-PAH is a rare complication of CTD, and the main type of CTD in our country is systemic lupus erythematosis, of which the patient population is mainly female. This retrospective cohort study is part of a larger prospective cohort study in our centre. Further researches with larger sample sizes from multi-centres to confirm the findings in this cohort study will be done in the future.
Comments 2: It looks like all the patients received PAH treatment. In the supplement they have mentioned monotherapy, dual therapy and triple therapy for PAH. They did not mention what kind of medications they received. In the world of PAH with limited treatment options, it will be interesting to know what they have received and if that played in any part in the 4 trajectories.
Response 2: Very nice comments which help us a lot. We agree that the specific type of PAH targeted drugs should be taken into concern. We have changed subgroups of PAH targeted drug therapy from ‘monotherapy, dual therapy, and triple therapy’ to ‘ERA, PDE5i, ERA+PDE5i, ERA+PRA, PDE5i+PRA, and ERA+PDE5i+PRA’. Unfortunately, PAH targeted drug therapy was still not significant in comparisons between trajectories, LASSO regression, and multivariate logistic regression. One possible reason is the small sample size of our cohort (trajectory analysis needs longitudinal data and many patients have not been followed-up for more than one year). Further researches focused on PAH treatment will be done in the future.
Comments 3: In the supplement table, its written that Intensive CTD immunotherapy was given in 39 out of the 52 patients. If you add up all patients in 4 groups its coming to 46 (11+19+7+9 = 46).
Response 3: We feel so sorry that the number of patients who received intensive CTD immunotherapy in the high stability trajectory is 2 (11+19+7+2 = 39), but not 9 (patients who received intensive CTD immunotherapy in the delayed remission and high stability trajectories is 9 (7+2 = 9)), and we have corrected it.
Reviewer 2 Report
Comments and Suggestions for Authors
Dear Authors,
Thank you for conducting this nice study.
I have a few concerns about the methods how the data were collected and stratified based on NT-proBNP levels and the interval of follow up. The duration of the disease and its severity were not considered.
The data analyses were insufficient and need further analyses with controlling for other covariate factors. Given that NT-proBNP levels are affected by other diseases, have you considered this during the analysis.
The discussion could be revised based on the new analyses.
Best wishes,
Comments on the Quality of English LanguageThe language could be improved by proofreading the manuscript.
Author Response
Thank you very much for taking the time to review this manuscript. Please find the detailed responses below and the corresponding revisions highlighted in the re-submitted files.
Comments 1: I have a few concerns about the methods how the data were collected and stratified based on NT-proBNP levels and the interval of follow up.
Response 1: We fully agree with your suggestions and we have improved the description in the materials and methods part. About the data collection: ‘Baseline data of demographics, detailed case histories, 6-minite walk distance, laboratory examinations, echocardiography, and RHC were obtained from electronic records systems and specialist services databases by clinicians blinded to the patient outcomes’. About the interval of follow up: ‘Patients were followed up every 3 months in outpatient clinic for re-evaluation including measurements of NT-proBNP levels in peripheral blood’. We should have mentioned that this retrospective cohort study is part of a larger prospective cohort study in our centre, so the interval of follow up was assigned as 3 months according to our study design.
Comments 2: The duration of the disease and its severity were not considered. The discussion could be revised based on the new analyses.
Response 2: Very nice comments which help us a lot. The CTD duration, the CTD activity, and the PAH duration had been considered while analyzing the association between baseline clinical characteristics and NT-proBNP trajectories. The PAH severity was reflected by echocardiographic and RHC measurements, and we add 6-minute walk distance and World Health Organization functional class as supplements according to your suggestions. The 2018 World Symposium on Pulmonary Hypertension (WSPH) simplified risk assessment scale risk groups were classified by experienced rheumatologists as a comprehensive evaluation of PAH severity, and its association with the NT-proBNP trajectories were analyzed independently to avoid severe collinearity.
Comments 3: The data analyses were insufficient and need further analyses with controlling for other covariate factors. Given that NT-proBNP levels are affected by other diseases, have you considered this during the analysis. The discussion could be revised based on the new analyses.
Response 3: Thank you very much for your comments. The first part of our study was to identify NT-proBNP trajectories and explore its association with prognosis. We considered NT-proBNP trajectories as comprehensive independent variables which were affected by a variety of factors including baseline factors and factors happened during the whole follow up. We did not adjust for covariate factors while analyzing between NT-proBNP trajectories and prognosis for the reason that NT-proBNP trajectories was considered as a result of all of the covariates. The second part of our study was to explore the association between baseline clinical characteristics and NT-proBNP trajectories. We assigned NT-proBNP trajectories as dependent variables. LASSO regression and multi-variate logistic regression were conducted so that covariate factors had been taken into concerns while analyzing between baseline clinical characteristics and NT-proBNP trajectories. We totally agree with the fact that NT-proBNP levels are affected by a variety of factors. We improved the description of the inclusion and the exclusion criteria. Patients with important interference factors of NT-proBNP levels such as age (> 75 years), eGFR (< 60 mL/min), left heart diseases (elevated left ventricular end-diastolic wall stress or atrial fibrillation) had been excluded according to the exclusion criteria before conducting all of the statistical analyses.
Comments 4: The language could be improved by proofreading the manuscript.
Response 4: Completely agree. English is not our native language. We asked for linguistic assistance from LetPub (www.letpub.com) after completing the first draft. We proofread our re-submitted manuscript repeatedly to reduce grammar error and improve the description. If necessary, we will ask for English editing from MDPI.
Reviewer 3 Report
Comments and Suggestions for Authors
The novelty and contribution of the approach in none or not well stated.
The approach and research methodology is poorly explained.
No justification behind GBTM and not proper explanation on the approach.
The findings are not compared with the state of the art.
Comparison with respect to other statistical measures such as information theoretic ones are required.
Author Response
Thank you very much for taking the time to review this manuscript. Please find the detailed responses below and the corresponding revisions highlighted in the re-submitted files.
Comments 1: The novelty and contribution of the approach in none or not well stated. No justification behind GBTM and not proper explanation on the approach.
Response 1: Thank you very much for your comments. Trajectory analysis based on longitudinal observational data could provide more information than analysis at single time point in the long-term management of patients. Methods of trajectory analysis include growth curve modeling (GCM), growth mixture modeling (GMM), group-based trajectory modeling (GBTM), and latent transition analysis (LTA). According to the inventors of GBTM, GBTM have a wide range of applications in clinical research, and has advantages in mapping the developmental course of trajectories over time, and identifying predictors of trajectory group membership. Compared with GMM, GBTM has lower computational complexity and fitting difficulty, which is more suitable for modeling with relatively less observation points. Therefore, we chose GBTM as the method of trajectory analysis in our study.
Comments 2: The approach and research methodology is poorly explained.
Response 2: We fully agree with your suggestions and we have improved the description in the materials and methods part. GBTM was conducted using SAS 9.4 (traj procedure). We generated models in which the number of trajectories ranged from three to four, and the shape of the trajectories was represented by polynomial functions ranging from zero to third order. The appropriate model was determined using the following criteria: (1) the fitting of each trajectory was statistically significant, (2) the average posterior probability (APP) of each trajectory was > 0.7, (3) the odds of correct classification (OCC) of each trajectory was > 5, (4) the relative entropy (Ej) was > 0.5, and (5) the number of individuals assigned to each trajectory exceeded 3% of the total population. We added a supplementary table and listed all of the eligible models (Table S1), and the final model was chosen based on its relatively small Bayesian information criterion (BIC) and clinical interpretability.
Comments 3: The findings are not compared with the state of the art.
Response 3: Thank you very much for your comments. Our first finding was the association between the NT-proBNP trajectories and the prognosis. There had been a few researchers on trajectory analyses of NT-proBNP in CTD-PAH patients. We had compared our finding with an existing study and the advantage of our study was that we had explored the effects of time to achieve NT-proBNP remission on prognosis. Our second finding was the effect of intensive CTD immunotherapy on NT-proBNP remission. To date, strategies for using CTD immunotherapy in CTD-PAH were rarely mentioned in authoritative guidelines, and only a few studies have investigated the effectiveness of CTD immunotherapy. We had compared our finding with the Chinese SLE treatment and research group (CSTAR)-PAH study and a systematic review derived from it. Intensive CTD immunotherapy was both beneficial in their cohort and ours. But the end point and the patient population were different.
Comments 4: Comparison with respect to other statistical measures such as information theoretic ones are required.
Response 4: We fully agree with your suggestions. We added a supplementary table and listed model evaluation parameters including BIC of all of the eligible models (Table S1). The final model was chosen based on its relatively small BIC and clinical interpretability.
Round 2
Reviewer 2 Report
Comments and Suggestions for Authors
Dear Authors,
Thank you for making the revision.
Best wishes,
Reviewer 3 Report
Comments and Suggestions for Authors
The questions are addressed well. The manusctipt is acceptable.